# Gestational Hypertension and High-Density Lipoprotein Function: An Explorative Study in Overweight/Obese Women of the DALI Cohort

**DOI:** 10.3390/antiox12010068

**Published:** 2022-12-29

**Authors:** Julia T. Stadler, M. N. M. van Poppel, Christina Christoffersen, David Hill, Christian Wadsack, David Simmons, Gernot Desoye, Gunther Marsche

**Affiliations:** 1Division of Pharmacology, Otto Loewi Research Center for Vascular Biology, Immunology and Inflammation, Medical University of Graz, Universitätsplatz 4, 8010 Graz, Austria; 2Institute of Human Movement Science, Sport and Health, University of Graz, 8010 Graz, Austria; 3Department of Biomedical Science, University of Copenhagen, 2200 Copenhagen, Denmark; 4Department of Biochemistry, Rigshospitalet, University Hospital of Copenhagen, 2200 Copenhagen, Denmark; 5Lawson Health Research Institute, London, ON N6C 2R5, Canada; 6Research Unit, Department of Obstetrics and Gynecology, Medical University of Graz, 8036 Graz, Austria; 7BioTechMed-Graz, 8010 Graz, Austria; 8Macarthur Clinical School, Western Sydney University, Sydney, NSW 2560, Australia

**Keywords:** gestational hypertension, HDL, cholesterol efflux capacity, paraoxonase-1, apolipoprotein M, pregnancy, obesity

## Abstract

Gestational hypertension (GHTN) is associated with an increased cardiovascular risk for mothers and their offspring later in life. High-density lipoproteins (HDL) are anti-atherogenic by promoting efflux of cholesterol from macrophages and suppression of endothelial cell activation. Functional impairment of HDL in GHTN-complicated pregnancies may affect long-term health of both mothers and offspring. We studied functional parameters of maternal and neonatal HDL in 192 obese women (pre-pregnancy BMI ≥ 29), who were at high risk for GHTN. Maternal blood samples were collected longitudinally at <20 weeks, at 24–28 and 35–37 weeks of gestation. Venous cord blood was collected immediately after birth. Maternal and cord blood were used to determine functional parameters of HDL, such as HDL cholesterol efflux capacity, activity of the vaso-protective HDL-associated enzyme paraoxonase-1, and levels of the HDL-associated anti-inflammatory apolipoprotein (apo)M. In addition, we determined serum anti-oxidative capacity. Thirteen percent of the women were diagnosed with GHTN. While we found no changes in measures of HDL function in mothers with GHTN, we observed impaired HDL cholesterol efflux capacity and paraoxonase-1 activity in cord blood, while serum antioxidant capacity was increased. Of particular interest, increased maternal paraoxonase-1 activity and apoM levels in early pregnancy were associated with the risk of developing GHTN. GHTN significantly impairs HDL cholesterol efflux capacity as well as HDL PON1 activity in cord blood and could affect vascular health in offspring. Maternal paraoxonase-1 activity and apoM levels in early pregnancy associate with the risk of developing GHTN.

## 1. Introduction

During pregnancy, various metabolic, vascular and physiological changes occur to ensure a continuous supply of essential nutrients for fetal growth and development [1]. The physiological stress of pregnancy can also lead to the development of adverse maternal pregnancy outcomes, such as gestational hypertension (GHTN). GHTN is defined as systolic blood pressure > 140 mmHg and diastolic blood pressure > 90 mmHg. According to the WHO, this pregnancy disorder is one of the main causes of maternal and fetal morbidity, and one of the most common causes of maternal death in Europe [2,3]. Obesity is one of the main risk factors for developing GHTN and the more severe condition of preeclampsia [4]. This is of particular importance given the increasing prevalence of obesity, which is a serious public health problem associated with the lifestyle prevalent in the developed world [5]. Hypertensive pregnancy disorders have public health implications that extend far beyond affected pregnancies, as GHTN is associated with an increased risk of cardiovascular complications later in life for mothers and their offspring [6,7,8].

Serum levels of high-density lipoprotein (HDL)-cholesterol (HDL-C) are inversely associated with the risk of developing cardiovascular disease. HDL particles promote reverse cholesterol transport, i.e., the uptake of excessive cholesterol from peripheral cells and its delivery to the liver for excretion. This cholesterol efflux capacity of HDL has been shown to be inversely associated with the incidence of cardiovascular events, independent of HDL-C levels [9]. Lipoproteins not only serve as lipid transporters but are also known to exert important anti-inflammatory and immunomodulatory functions [10]. HDL exhibits vascular-protective activities and anti-oxidative and anti-inflammatory functions through HDL-associated enzymes and apolipoproteins [11,12,13]. The bioactive lipid sphingosine-1-phosphate (S1P) is mainly bound to apolipoprotein M (apoM) of HDL and mediates many beneficial effects in hypertension and cardiac hypertrophy on the vasculature via G protein–coupled S1P receptors [14,15]. Moreover, HDL associated paraoxonase-1 (PON1) shows anti-inflammatory properties and is an important determinant for the capacity of HDL to stimulate endothelial nitric oxide production [16].

An increasing number of studies have demonstrated a significant effect of inflammatory disorders on HDL composition and function [17,18,19,20]. Since pregnancy is a low-grade systemic inflammatory condition [21], changes in HDL structure occur and an increase of HDL particle size and changes in HDL protein composition were reported in pregnant women (18–24 weeks of gestation) [22].

We hypothesized that HDL functionality would be impaired in women diagnosed with GHTN and in their offspring. In this longitudinal study, obese women considered at high risk for pregnancy complications were followed from their recruitment early in pregnancy until delivery. We assessed several key functions of HDL, including the ability to remove cholesterol from macrophages (cholesterol efflux capacity), the activity of the HDL-associated enzyme paraoxonase-1 (PON1), an vaso-protective anti-inflammatory enzyme [23], and serum levels of HDL associated anti-inflammatory apolipoprotein M (apoM) [24]. In addition, we assessed neonatal and maternal serum total anti-oxidative capacity. We further investigated whether HDL function in early pregnancy is related to GHTN risk.

## 2. Methods

This is a secondary analysis of the vitamin D And Lifestyle Intervention for gestational diabetes mellitus prevention (DALI) study, a randomized controlled trial (ISRCTN70595832) [25,26]. Further details including recruitment and inclusion criteria are provided in the Appendix A. All local ethics committees provided ethical approval and written informed consent was signed by all participants prior to data collection. Maternal blood samples were longitudinally collected <20 weeks, 24–28 and 35–37 weeks of gestation, and venous cord blood was collected immediately after birth.

### 2.1. Definition of GHTN

Information on GHTN was collected from medical records. GHTN was defined as systolic blood pressure > 140 mmHg and/or diastolic blood pressure > 90 mmHg without proteinuria after 20 weeks of gestation.

### 2.2. Biochemical Analyses

Glucose was measured using the hexokinase method (DiaSys Diagnostic Systems, Holzheim, Germany) with a lower limit of sensitivity of 0.1 mmol/L.

Insulin was quantified by a sandwich-immunoassay (ADVIA Centaur, Siemens Healthcare Diagnostics Inc., Vienna, Austria) with an analytical sensitivity of 0.5 mU/L, intra-assay CVs of 3.3–4.6% and inter-assay CVs of 2.6–5.9%. All assays were carried out following the instructions of the manufacturer. HOMA-IR was calculated as [glucose*insulin]/22.5 mmol/L*UI/mL.

Total cholesterol (cord blood only) and triglycerides, were measured using colorimetric enzymatic assays using reagents from DiaSys Diagnostic Systems (Holzheim, Germany) and were calibrated using secondary standards from Roche Diagnostics (Mannheim, Germany). HDL-C was measured with a homogenous assay from DiaSys Diagnostics, and LDL cholesterol (LDL-C) was calculated according to the Friedewald formula (LDL-C = TC − HDL-C − TG/5). Non-esterified fatty acids (FFAs) were quantified using an enzymatic reagent and standards from Wako Chemicals (Neuss, Germany). All lipid analyses were performed on an Olympus AU640 automatic analyzer (Beckman Coulter, Brea, CA, USA).

### 2.3. ApoB-Depletion of Serum

For the analyses of HDL composition and function, serum HDL (apoB-depleted serum) was used. Polyethylene glycol (Sigma Aldrich, Darmstadt, Germany) (40 µL; 20% in 200 mmol/L glycine buffer) was added to 100 µL serum, mixed gently, and then incubated for 20 min at room temperature. After a centrifugation step at 10,100× *g* for 30 min at 4 °C, the supernatant was collected. Samples were stored at −70 °C until use.

### 2.4. Cholesterol Efflux Capacity

The cholesterol efflux capacity of apoB-depleted serum was assessed as described previously [27]. In brief, J774.2 macrophages (Sigma-Aldrich, Darmstadt, Germany) were maintained in DMEM media (containing 10% FBS, 1% PS). Cells were seeded on 48-well plates (300,000 cells/well), cultured for 24 h and loaded with 0.5 µCi/mL radiolabelled [^3^H]-cholesterol in medium containing 2% FBS, 1% PS and 0.3 mM 8-(4-chlorophenylthio)-cyclic AMP overnight. After that, cells were washed and equilibrated in serum-free media supplemented with 0.2% BSA for 2 h. To determine [^3^H]-cholesterol efflux, cells were incubated with 2.8% apoB-depleted serum for 3 h at 37 °C. Cholesterol efflux capacity was expressed as radioactivity in cell culture supernatant relative to total radioactivity in cells and supernatant.

### 2.5. Arylesterase (AE)—Activity of Paraoxonase1

The Ca^2+^-dependent AE-activity of PON1 was determined with a photometric assay using the substrate phenylacetate as described elsewhere [28]. Briefly, apoB-depleted serum was diluted 10-fold and 1.5 µL were added to 200 µL reaction buffer (100 mM Tris, 2 mM CaCl_2_, 1 mM phenylacetate). The rate of hydrolysis of the substrate was monitored by the increase of absorbance at 270 nm.

### 2.6. Anti-Oxidative (AO)—Capacity of ApoB-Depleted Serum

As previously described [28], the AO-capacity of apoB-depleted serum was assessed with a fluorometric assay using the fluorescent dye dihydrorhodamine. The dye was suspended in DMSO (50 mM stock), which was diluted in HEPES (20 mM HEPES, 150 mM NaCl_2_, pH 7.4) containing 1 mM 2,2′-azobis-2-methyl-propanimidamide-dihydrochloride (AAPH) to a 10 μM working reagent. In a 384-well plate, 10 µL apoB-depleted serum dilution (1:10) were placed and 90 µL working reagent was added. The increase in fluorescence due to oxidation of dihydro-rhodamine was monitored for 90 min at 538 nm. The increase in dihydro-rhodamine fluorescence per minute in the absence of apoB-depleted serum was set at 100%, and individual apoB-depleted serum samples were calculated as the percentage of inhibition of dihydro-rhodamine oxidation.

### 2.7. Serum Levels of Apolipoprotein M (apoM)

Quantification of serum apoM was performed using a sandwich ELISA based assay as previously described [29].

### 2.8. Statistical Analyses

Participant characteristics are presented by mean and standard deviation (SD), median and interquartile range (IQR) or count and proportion. Maternal and neonatal characteristics were compared between the groups of women with and without GHTN using unpaired *t*-test and chi-square tests. Characteristics of included and excluded participants were compared using unpaired *t*-test and chi-square test. Differences in HDL parameters between time points were tested by paired sample *t*-tests.

The differences in HDL-related parameters between women with GHTN and women without this complication were tested using Student’s *t* test (maternal samples) or Mann–Whitney U test (cord blood samples). We performed logistic regression analysis to identify possible associations between HDL-related parameters measured at baseline (<20 weeks) and the pregnancy complication GHTN. For better comparison, z-scores of the HDL-related parameters were used in the models. This means that the ORs are the chance of developing GHTN with one SD increase in the HDL-related parameter. Models were adjusted for maternal age, BMI and HOMA-R. In sensitivity analyses, possible confounding by maternal smoking, intervention allocation, or (for cord blood HDL parameters) mode of delivery was assessed.

To assess the association of GHTN with functional parameters of HDL in cord blood, we performed linear regression analyses, adjusted for maternal age, parity and gestational age at birth.

All analyses were performed in IBM SPSS (Version 27.0. Armonk, NY, USA: IBM Corp). A *p* value of <0.05 was used for determining statistical significance.

## 3. Results

### 3.1. Study Cohort Characteristics

A description of study cohort characteristics is provided in Table 1. Participants were selected based on the availability of serum samples from all time points of pregnancy (<20 weeks, 24–28, and 35–37 weeks of gestation; *n* = 192). As information on GHTN prevalence was not available from all women, 185 participants were selected for the analyses between GHTN and no GHTN. Women included in these analyses were higher educated, smoked less frequently, and their neonates had a higher birth weight compared to the total DALI study population in the pilot and lifestyle trials (Appendix A). During pregnancy, 24 (13%) women were diagnosed with GHTN. There were no significant differences in maternal or neonatal characteristics between women with or without GHTN.

### 3.2. Changes of HDL-Related Parameters during Pregnancy

Maternal serum levels of HDL-C increased modestly in our study between <20 weeks and 24–28 weeks. Interestingly, all HDL-related parameters, besides HDL-C, changed from early to late pregnancy, with a significant increase in the HDL cholesterol efflux capacity and serum apoM levels, whereas PON1 activity decreased from <20 weeks to 35–37 weeks of pregnancy. In addition, the anti-oxidative capacity of serum was increased (Figure 1). After correction for differences in HDL-C levels (by normalization to HDL-C), no differences in parameters of HDL function were observed between the time points of pregnancy (Appendix A).

### 3.3. HDL-Related Parameters in Cord Blood

Compared to maternal serum, anti-oxidative capacity of serum was higher in circuit of the offspring, while PON1 activity, cholesterol efflux capacity and serum apoM levels were markedly lower compared to maternal levels (Figure 1). To determine the functionality of individual HDL particles, we normalized the measured functional parameters to HDL-C levels (Appendix A). Compared with mothers, the HDL cholesterol efflux capacity of individual HDL particles in cord blood was significantly higher, indicating differences in the structure and composition of fetal HDL. ApoM content of HDL particles in cord blood was significantly increased (Appendix A). ApoM improves cholesterol efflux capacity of HDL [30,31], consistent with the increased cholesterol efflux capacity of individual HDL particles in cord blood. Interestingly, PON1 activity of individual HDL particles was lower in cord blood (Appendix A). We observed no sex differences in HDL-related parameters in the offspring cohort.

### 3.4. GHTN-Associated Changes in HDL-Related Parameters in Mothers and Cord Blood

Next, we assessed parameters of HDL function in women diagnosed with GHTN and their offspring.

We observed that GHTN was not associated with altered maternal HDL-C, HDL cholesterol efflux capacity, PON1 activity or anti-oxidative capacity (Figure 2A–D). Of particular interest, in cord blood, HDL cholesterol efflux capacity and PON1 activity were significantly impaired, whereas HDL-C levels were unaltered (Figure 2A–C). Moreover, we observed an increased anti-oxidative capacity of cord serum (Figure 2D) (*p* = 0.04). Serum apoM levels showed a non-significant trend toward higher levels in women diagnosed with GHTN at all-time points of pregnancy but lower levels in offspring (Figure 2E).

### 3.5. Association of HDL-Related Parameters with Pregnancy Outcome

We performed logistic regression analysis to identify possible associations between HDL-related parameters measured at baseline (<20 weeks) and GHTN (Figure 3). The models were adjusted for maternal age, BMI and HOMA-R. Interestingly, women having increased serum PON1 activity as well as increased apoM levels were at a higher risk of developing GHTN. Serum levels of HDL-C, cholesterol efflux capacity and the serum anti-oxidative capacity were not associated with GHTN incidence. Further adjustments for maternal smoking or interventions did not substantially change the results.

### 3.6. Consequences of Maternal Pregnancy Disorders on HDL-Related Parameters in Cord Blood

We then conducted linear regression analyses to investigate the relationship between GHTN and the functional parameters of HDL in cord blood (Table 2). Cord blood was available from 102 pregnancies, and the model was adjusted for maternal age, parity, and gestational age at birth. Of particular interest, maternal GHTN was associated with a decrease in PON1 activity and cholesterol efflux capacity and an increase in anti-oxidative capacity of cord serum. No association was found between GHTN and HDL-C levels. Further adjustment for maternal smoking, intervention or mode of delivery did not change the results.

## 4. Discussion

In the current study, we assessed GHTN-associated changes in HDL-related functional parameters in mothers and their offspring. We observed that GHTN was associated with marked changes in HDL function in the offspring, whereas HDL functionality was not affected in mothers. Specifically, GHTN was associated with impaired HDL cholesterol efflux capacity and PON1 activity in cord blood, whereas serum anti-oxidative capacity was increased.

It is well known that during the course of pregnancy, HDL-C levels increase modestly in mothers [32,33,34], which was also seen in our study from <20 weeks to 24–28 weeks of gestation. We observed increased maternal HDL cholesterol efflux capacity, apoM levels and anti-oxidative capacity of serum during the course of pregnancy. However, after normalization of functional HDL parameters to HDL-C levels, no significant differences remained, suggesting that the functionality of individual HDL particles was not altered during the course of pregnancy.

Pregnancy induced changes in HDL function in gestational hypertension have remained poorly investigated. A few previous studies have focused on analysing changes of HDL structure and function in women with preeclampsia, a more severe form of GHTN. HDL of mothers diagnosed with preeclampsia showed an increased particle diameter and reduced PON1 activity and was less effective in reducing adhesion molecule expression on endothelial cells [35]. Mixed results were reported when HDL cholesterol efflux capacities were assessed in preeclampsia. One study reported increased maternal and fetal HDL cholesterol efflux capacities in preeclampsia [36], whereas another study reported that women with a history of preeclampsia (6 months postpartum) display decreased HDL cholesterol efflux capacity [37]. These data suggest that in more severe forms of GHTN, parameters of HDL function in maternal blood may also be affected.

A very interesting finding of our study was that maternal PON1 activity and apoM levels in early pregnancy were associated with the risk of developing GHTN. PON1 is capable of hydrolyzing a wide spectrum of substrates including oxidized lipids and is thought to play a role in the development of a large variety of diseases with an inflammatory component, including heart disease, diabetes, rheumatic diseases, neurological diseases and cancer [38,39]. Given this close relationship between PON1 activity and various diseases, further studies would be of great interest to investigate the extent to which PON1 activity can serve as an early marker for a wide variety of pregnancy-related diseases.

ApoM has gained attention in recent years, as studies have shown that apoM is important for the formation of preβ-HDL increasing cholesterol efflux capacity and promoting endothelial protective activities via its bound S1P [15,40,41]. Perinatal and long-term offspring morbidities are strongly dependent on the preservation of placental vascular homeostasis during pregnancy. Recent studies have shown that impaired S1P signalling in the endothelium indicates the health/disease state of the vasculature and is thought to contribute to the pathogenesis of preeclampsia [14,42]. Thus, a decrease in apoM in GHTN could have negative consequences for endothelial function in neonates with long-term pathological implications for the heart later in life.

An important finding of our study was that GHTN significantly impaired HDL cholesterol efflux capacity as well as HDL PON1 activity in cord blood. Both factors could increase long-term cardiovascular risk of the offspring. This notion is supported by recent studies showing that reduced HDL cholesterol efflux capacity predicts cardiovascular risk independent of HDL-C [9] and that low PON1 activity is linked to systemic oxidative stress and prospective cardiovascular risk [43].

On the other hand, surprisingly, GHTN was associated with increased antioxidant capacity of cord blood serum. This might indicate a compensatory mechanism in response to reduced blood flow and increased oxidative stress in the neonatal/placental circulation. It is important to note that the anti-oxidative capacity of serum is predominantly determined by serum albumin levels in addition to low-molecular-weight antioxidants with a minor contribution of HDL [44]. It is reasonable to speculate that decreased excretion of hydrophilic antioxidants may explain the increased antioxidant capacity of cord blood serum.

Some limitations of our study have to be acknowledged. A limitation is the small sample size of GHTN patients. Therefore, further larger studies are needed to confirm our results and to draw firm conclusions. Because this study was originally designed as an intervention study for the prevention of GDM in obese women, no lean control group was available. Moreover, due to the small sample size of GHTN women, interactions with offspring sex were not assessed, and due to very low prevalence of women with preeclampsia, these participants were not included in our analyses.

Strengths of our study are that we assessed multiple functional parameters of HDL and its prospective and longitudinal study design. Maternal serum samples were collected at three time-points during pregnancy, which enabled us to study changes overtime. Moreover, paired mother-offspring blood samples available for all pregnancies included. To the best of our knowledge, this is the largest study on effects of obesity and pregnancy disorders on HDL-related parameters in mothers and offspring. Moreover, it is a pan-EU study, which is representative of pregnant Caucasian women with obesity in Europe, who are well phenotyped.

## 5. Conclusions

In this study, we demonstrated that HDL-C and functional parameters of HDL change over the duration of pregnancy. In addition, we showed that GHTN does not significantly alter maternal HDL-related parameters, but has profound effects on HDL-related parameters in the offspring. Follow-up studies are needed to clarify when and whether these changes normalize after birth or whether the changes contribute to long-term cardiovascular risk of the offspring. In addition, our results suggest that maternal PON1 activity and apoM levels are associated with the risk of developing GHTN.

## Figures and Tables

**Figure 1 antioxidants-12-00068-f001:**
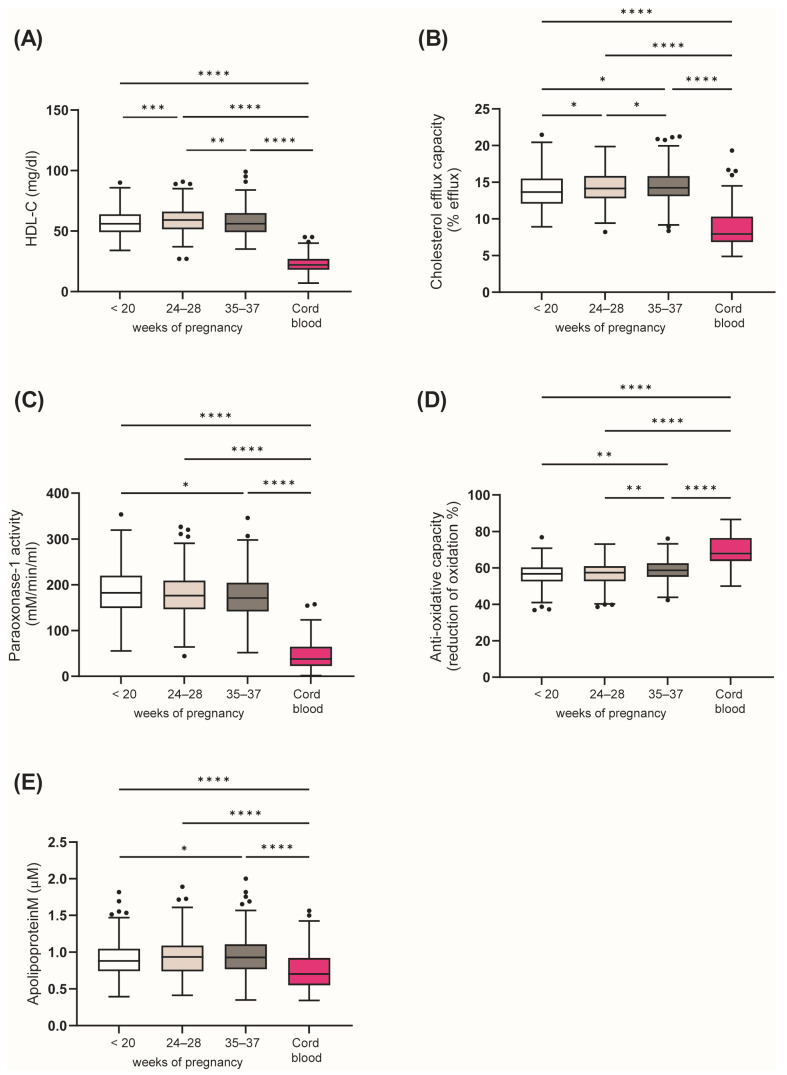
Changes in HDL-related parameters during pregnancy and comparisons with cord blood. HDL-C (**A**), the ability to promote cholesterol efflux (**B**) and activity of HDL-associated PON1 (**C**) were assessed. (**D**) Serum anti-oxidative capacity and (**E**) apolipoprotein M levels. Data are presented as Tukey-Boxplots, showing the median and interquartile ranges, as well as minimum, maximum values and outliers. * *p* < 0.05, ** *p* < 0.01, *** *p* < 0.001, **** *p* < 0.0001 based on paired *t*-test.

**Figure 2 antioxidants-12-00068-f002:**
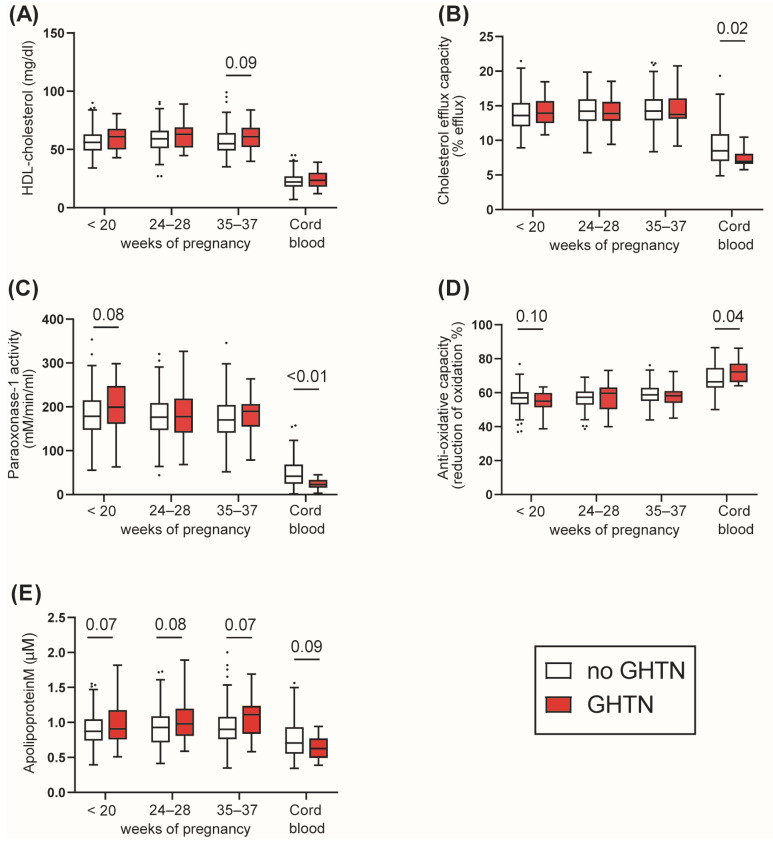
Differences in HDL-related parameters in women with GHTN and healthy controls. HDL-cholesterol (**A**), cholesterol efflux capacity (**B**) and the activity of HDL-associated PON1 (**C**) were assessed. (**D**) Serum anti-oxidative capacity and (**E**) apolipoprotein M levels. Data are presented as Tukey-Boxplots showing the median and interquartile ranges as well as minimum, maximum values and outliers. Significant (*p* < 0.05) and non-significant trends (*p* < 0.10) are indicated with *p*-value. Differences were analysed by student *T*-test (maternal samples) or Mann–Whitney U test (cord blood).

**Figure 3 antioxidants-12-00068-f003:**
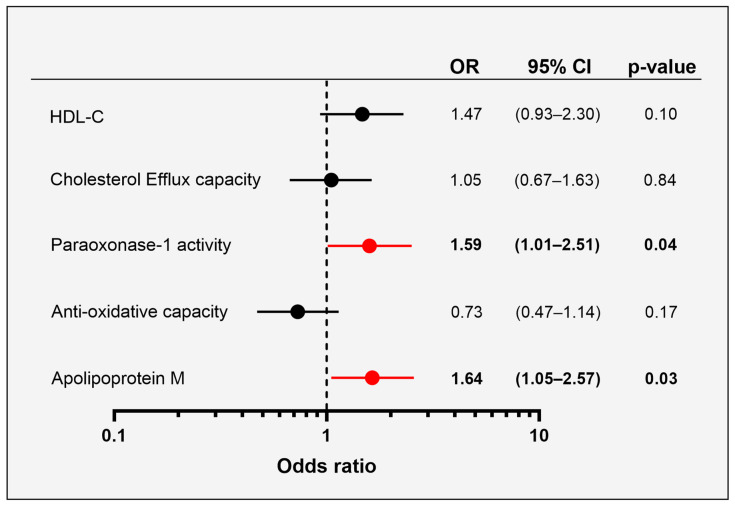
Forest plot with logistic regression analyses showing associations between HDL-related parameters (z-scores) at <20 weeks’ gestation and the risk of developing GHTN in the second and third trimester of pregnancy. Odds ratio per 1 SD increase of the variables. Models were adjusted for maternal age, BMI and HOMA-R. Significant results are highlighted in red.

**Table 1 antioxidants-12-00068-t001:** Characteristics of study cohort.

Maternal Characteristics	Total *n* = 192	No GHTN *n* = 161	GHTN *n* = 24	*p*
Age, years	31.7 ± 5.4	31.9 ± 5.3	30.4 ± 5.3	0.19
Prepregnancy BMI, kg/m^2^	34.1 ± 4.5	33.9 ± 4.3	36.2 ± 5.8	0.07
Gestational weight gain, kg	8.4 ± 4.8	8.3 ± 5.0	8.9 ± 4.0	0.59
Primiparous	102 (53%)	82 (51%)	16 (67%)	0.15
Married/living with partner	181 (94%)	151 (94%)	23 (96%)	0.69
High education	119 (62%)	103 (64%)	14 (58%)	0.59
European descent	170 (89%)	142 (88%)	22 (92%)	0.62
Smoking	18 (9%)	17 (11%)	0 (0%)	0.09
GDM	55 (29%)	44 (28%)	7 (29%)	0.87
GHTN	24 (13%)	--	--	--
Preeclampsia	6 (3%)	5 (3%)	1 (4%)	0.78
Neonatal characteristics				
Gestational age at birth	39.7 ± 1.4	39.7 ± 1.4	39.5 ± 1.3	0.50
Birthweight	3575 ± 520	3587 ± 527	3521 ± 462	0.56
Female sex	90 (47%)	74 (46%)	15 (63%)	0.13

**Table 2 antioxidants-12-00068-t002:** Linear regression models of the association of GHTN with cord blood HDL-parameters.

	HDL-C B (95% CI)	*p*	Anti-Oxidative Capacity B (95% CI)	*p*	Paraoxonase-1 Activity B (95% CI)	*p*	Cholesterol Efflux Capacity B (95% CI)	*p*	ApoM B (95% CI)	*p*
GHTN	0.04 (−0.06; 0.14)	0.44	5.29 (0.63; 9.95)	0.03	−24.88 (−41.87; −7.89)	0.005	−2.04 (−3.61; −0.47)	0.01	−0.14 (−0.28; 0.003)	0.055

Models adjusted for maternal age, parity and gestational age at birth.

## Data Availability

The raw data supporting the conclusions of this manuscript will be made available by the authors, without undue reservation, on request to the corresponding author.

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
