# Peer review of "Gestational Hypertension and High-Density Lipoprotein Function: An Explorative Study in Overweight/Obese Women of the DALI Cohort"

_antioxidants, 2022, doi:10.3390/antiox12010068_

Round 1

Reviewer 1 Report

Manuscript ID: antioxidants-2056995

Title: Gestational hypertension and high-density lipoprotein function: An explorative study in overweight/obese women of the DALI cohort 

This study aimed to investigate GHTN-associated changes in HDL-related functional parameters in mothers and their offspring. Maternal and cord blood were used to determine multiple parameters. Their findings showed that GHTN was associated with significantly changes in HDL function in the offspring. The authors suggested that PON1 and apoM may serve as predictors for early identification of women at risk for developing GHTN. The topic described in this manuscript is interesting. However, the limitation of this study is small sample size (GHTN n=24). The sample size is too small to draw conclusions.

Author Response

Reviewer 1: This study aimed to investigate GHTN-associated changes in HDL-related functional parameters in mothers and their offspring. Maternal and cord blood were used to determine multiple parameters. Their findings showed that GHTN was associated with significantly changes in HDL function in the offspring. The authors suggested that PON1 and apoM may serve as predictors for early identification of women at risk for developing GHTN. The topic described in this manuscript is interesting. However, the limitation of this study is small sample size (GHTN n=24). The sample size is too small to draw conclusions.

Answer: We are pleased that our manuscript was positively received by the reviewer and are happy to consider the helpful comments.

We agree with the reviewer that the small sample size of women diagnosed with GHTN is a limitation of our study. According to the reviewer’s suggestion, we have deleted statements regarding prediction of GHTN in the manuscript and have addressed the small sample size in the limitations section. We have added the sentence “A limitation is the small sample size of GHTN patients. Therefore, further larger studies are needed to confirm our results and to draw firm conclusions” (line 338).

Reviewer 2 Report

This is very well-designed study and equally well-written manuscript. I concur with the authors about the limitations stated. However, due to the nature of the longitudinal study design, some of the limitations can be accommodated. The results are very clear and the statistical analyses and references from the data reflect the authors' conclusions. I support a decision to accept this manuscript for publication.

Author Response

Reviewer 2: This is very well-designed study and equally well-written manuscript. I concur with the authors about the limitations stated. However, due to the nature of the longitudinal study design, some of the limitations can be accommodated. The results are very clear and the statistical analyses and references from the data reflect the authors' conclusions. I support a decision to accept this manuscript for publication.

 Answer: We are pleased that our manuscript was favorably received by the reviewer and thank the reviewer for his positive comments.

Round 2

Reviewer 1 Report

No further comments